# DBD Plasma Combined with Different Foam Metal Electrodes for CO_2_ Decomposition: Experimental Results and DFT Validations

**DOI:** 10.3390/nano9111595

**Published:** 2019-11-11

**Authors:** Ju Li, Xingwu Zhai, Cunhua Ma, Shengjie Zhu, Feng Yu, Bin Dai, Guixian Ge, Dezheng Yang

**Affiliations:** 1Key Laboratory for Green Processing of Chemical Engineering of Xinjiang Bingtuan, School of Chemistry and Chemical Engineering, Shihezi University, Shihezi 832003, China; leej222@163.com (J.L.); zsj497262724@gmail.com (S.Z.); yufeng05@mail.ipc.ac.cn (F.Y.); db_tea@shzu.edu.cn (B.D.); 2Key Laboratory of Ecophysics, College of Sciences, Shihezi University, Shihezi 832003, China; 3Laboratory of Plasma Physical Chemistry, School of Physics, Dalian University of Technology, Dalian 116024, China

**Keywords:** dielectric barrier discharge plasma, foam metal electrodes, CO_2_ decomposition, density functional theory

## Abstract

In the last few years, due to the large amount of greenhouse gas emissions causing environmental issue like global warming, methods for the full consumption and utilization of greenhouse gases such as carbon dioxide (CO_2_) have attracted great attention. In this study, a packed-bed dielectric barrier discharge (DBD) coaxial reactor has been developed and applied to split CO_2_ into industrial fuel carbon monoxide (CO). Different packing materials (foam Fe, Al, and Ti) were placed into the discharge gap of the DBD reactor, and then CO_2_ conversion was investigated. The effects of power, flow velocity, and other discharge characteristics of CO_2_ conversion were studied to understand the influence of the filling catalysts on CO_2_ splitting. Experimental results showed that the filling of foam metals in the reactor caused changes in discharge characteristics and discharge patterns, from the original filamentary discharge to the current filamentary discharge as well as surface discharge. Compared with the maximum CO_2_ conversion of 21.15% and energy efficiency of 3.92% in the reaction tube without the foam metal materials, a maximum CO_2_ decomposition rate of 44.84%, 44.02%, and 46.61% and energy efficiency of 6.86%, 6.19%, and 8.85% were obtained in the reaction tubes packed with foam Fe, Al, and Ti, respectively. The CO_2_ conversion rate for reaction tubes filled with the foam metal materials was clearly enhanced compared to the non-packed tubes. It could be seen that the foam Ti had the best CO_2_ decomposition rate among the three foam metals. Furthermore, we used density functional theory to further verify the experimental results. The results indicated that CO_2_ adsorption had a lower activation energy barrier on the foam Ti surface. The theoretical calculation was consistent with the experimental results, which better explain the mechanism of CO_2_ decomposition.

## 1. Introduction

Recently, the environmental impact of carbon dioxide (CO_2_) has been extensively studied. The contribution of CO_2_ to the greenhouse effect has been confirmed, so solutions to the problem of excessive CO_2_ need to be found. Nowadays, the greenhouse effect originated from CO_2_ emission is one of the main challenges for human beings. Meanwhile, CO_2_ is a ubiquitous and universally available C1 (a substance containing one carbon atom) resource in the world. In this regard, three major strategies have been proposed: carbon capture, storage, and utilization [1,2,3,4,5]. As one of the carbon utilization pathways, CO_2_ direct decomposition has aroused special interest, because it can convert the greenhouse gas into value-added carbon monoxide (CO), which can be used not only as a fuel, but also as a widely used chemical raw material [6].

CO_2_ thermal decomposition requires temperature excess of 2000 K to form CO and O_2_ [7]. Hence, this is a high consumption energy process. In recent years, some alternative processes for CO_2_ decomposition have been developed, among them non-thermal plasma (NTP) is an emerging alternative to CO_2_ decomposition that has many merits, such as exciting high-energy electrons for reaction under environmental conditions. Meanwhile, the gas temperature of NTP can be very low, which ensures a low energy cost due to reduced heat loss [8]. Therefore, various NTP configurations have been tested, such as corona discharge [9], glow discharge [10], gliding arc discharge [11], microwave discharge [12], and radio frequency discharge [13]. In the near future, dielectric barrier discharge (DBD) plasma will be used in CO_2_ conversion [14].

DBD is one of the NTPs, which can replace the catalytic chemical process under high-temperature operating conditions. There are numerous microdischarges in the DBD reaction tube. Microdischarges are composed of a number of small current filaments, which are generally short-lived, only a few nanoseconds, and evenly distributed around the high-voltage electrode [15]. In a DBD, the average temperature of energetic electrons is very high, over the range of 10,000–100,000 K, but the actual gas temperature is close to the environment temperature [16]. Except for the generation of high-energy particles, DBD also produces ultraviolet-visible light, ozone species, excited species, radicals, ions, etc. These reactive species are responsible for the efficient initiation and propagation of reactions [17,18].

In general, the DBD reactor performance depends on reactor configuration, flow velocity, power, ambient gas, and catalyst/packing material [19,20]. Hueso et al. [21] conducted a DBD technique to reform methane and directly decompose methanol under normal and low temperature conditions. Depending on the applied voltage, feed ratio, reactants’ residence time, or reactor configuration, the conversion can reach 20–80% in the case of methane and 7–45% for CO_2_. Under similar experimental conditions, methanol direct decomposition is up to 60–100%. Tu et al. [22] observed shifts in CO_2_ conversion and process energy efficiency by changing DBD plasma processing parameters. Conclusion can be made that increasing the discharge power or lowering the gas flow velocity can improve CO_2_ conversion with other parameters unchanged, but with lower energy efficiency. Snoeckx et al. [23] conducted extensive and in-depth studies on DBD plasma decomposition of CO_2_. It can be found that the presence of N_2_ in 50% of N_2_ gas mixture has little effect on CO_2_ conversion and energy efficiency. However, a higher N_2_ fraction results in a decrease in CO_2_ conversion and energy efficiency. Michielsen et al. [24] reported CO_2_ decomposition in a packed-bed DBD reactor packed with glass wool, quartz wool, and SiO_2_, ZrO_2_, Al_2_O_3_, and BaTiO_3_ spherical beads of different sizes. Among the many filler materials studied, BaTiO_3_ has a maximum conversion of up to 25% and an energy efficiency of 4.5%. Uytdenhouwen et al. [25] used a DBD microplasma reactor to investigate the effect of gap size reduction combined with packing material on the conversion and efficiency of CO_2_ dissociation. The results were compared to a conventional size reactor as a reference. Even though the energy efficiency is low, decreasing the discharge gap can greatly enhance the CO_2_ conversion rate.

At present, in the literatures we have seen, most of the catalysts used for CO_2_ decomposition by plasma are metal oxides. However, a few literatures have reported researches on CO_2_ conversion by using foam metals as catalysts. In this work, we studied DBD plasma junctions and different packed foam metal materials (Fe, Al, and Ti) for CO_2_ conversion. Compared to metal oxide catalysts, the foam metals can not only serve as a carrier for energy transformation, but the foam metals distributed in the discharge gap also consume a part of O_2_ and O radicals, so that the reaction proceeds in the positive direction, promoting the decomposition of CO_2_. The effects of power, flow velocity, and other discharge characteristics are studied to better understand the influence of the filling catalysts on CO_2_ splitting. Furthermore, we use density functional theory (DFT) to verify the experimental results. The results show that CO_2_ adsorption has a lower activation energy barrier on the foam Ti surface. 

## 2. Experimental

### 2.1. Dielectric Materials

In this paper, foam metals (Fe, Al, and Ti) are used as filler catalysts along the discharge region. All of these catalysts are commercially available and untreated.

### 2.2. Experimental Setup

The schematic diagram of an experimental device for CO_2_ splitting by DBD plasma is shown in Figure 1. The experimental device includes the following components: plasma generator, DBD reactor (Beijing Synthware glass), and a gas chromatograph (GC) for detecting the product. The coaxial cylindrical reactor has two electrodes, namely a high-voltage (HV) electrode and a low-voltage (LV) electrode, and they produce a discharge between the two electrodes. The HV electrode is a stainless steel rod, 2 mm in diameter, which is connected to an alternating current (AC) power source and fixed on the central shaft of the reaction tube. Condensed water at 20 °C is added to the reactor housing and the LV electrode is grounded with a wire. The discharge parameters were measured using a voltage probe, a current monitor, and an oscilloscope. The composition of the off-gas from the plasma reactor was determined by GC. The calculation of the discharge power is based on the area of the Lissajous figure displayed on the oscilloscope.

### 2.3. Gas Analysis and Parameter Definition

Pure CO_2_ (99.995%) is used as the reaction gas and the inlet feed gas flow velocity was varied from 20 to 100 mL/min through the mass flow controller. After the GC had stabilized, the composition of the gas products was recorded by GC every 15 min. Helium (He) was used as its carrier gas. 

CO_2_ conversion, CO yield, and selectivity are important parameters for evaluating the plasma process performance, as defined below:(1)$${\mathrm{CO}}_{2}\;\mathrm{conversion}\;\left(\%\right)=\frac{{\mathrm{CO}}_{2}\;\mathrm{converted}\;}{{\mathrm{CO}}_{2}\;\mathrm{input}}\times 100,$$
(2)$$\mathrm{CO}\;\mathrm{yield}\;\left(\%\right)=\frac{\mathrm{CO}\;\mathrm{formed}\;}{{\mathrm{CO}}_{2}\;\mathrm{input}}\times 100,$$
(3)$$\mathrm{CO}\;\mathrm{selectivity}\;\left(\%\right)=\frac{\mathrm{CO}\;\mathrm{formed}\;}{{\mathrm{CO}}_{2}\;\mathrm{converted}}\times 100.$$

Furthermore, specific input energy (SIE) is another important evaluation of system performance, defined as follows:(4)$$\mathrm{SIE}=\frac{\mathrm{discharge}\;\mathrm{power}\;}{{\mathrm{CO}}_{2}\;\mathrm{flow}\;\mathrm{rate}}\times 60.$$

The energy efficiency (EE) of the plasma splitting CO_2_ process is defined as the molar number of CO_2_ decomposed per unit plasma power and is formulated by the following equation:(5)$$\mathrm{EE}\;\left(\%\right)=\frac{\Delta H_{298K}\;\times {\mathrm{CO}}_{2}\;\mathrm{converted}\;}{\mathrm{SIE}\;\times 24.5}\times 100,$$
where ∆H represents the reaction enthalpy of one mole CO_2_ splitting.

## 3. Results and Discussion

### 3.1. Effect of Packing Materials on Discharge Characteristics

Figure 2 presents the variation of the Lissajous figure with discharge voltage. From the picture, we can observe that the area of the Lissajous figure increases with increasing voltage. The impact of different filled foam metals on the discharge characteristics at the same input power of 70 W is shown in Figure 3. Compared with non-packing materials, there are much less filamentary discharges in the reactor packed with foam Fe, Al, and Ti. It is apparent that the filling of foam metals in the reactor causes changes in discharge characteristics and discharge patterns, from the original filamentary discharge to both the current filamentary discharge and surface discharge. The results demonstrate that the introduction of the packed foam metals into the DBD reactor can promote the discharge characteristics. We have discussed similar phenomena in our previous studies. For example, when ZrO_2_ or foam Ni catalyst is introduced into the reactor, the discharge type changes and the filamentation discharge decreases dramatically [26,27,28]. Moreover, at the same input power, the Lissajous pattern of the electrode made of foam Ti has a larger area than the other two, which means that the foam Ti can further enhance the discharge power. Furthermore, from the slopes of the graph, it can be concluded that the foam Ti electrode transfers more charges than the other two foam metals. Filling the materials into the tube can generate a strong electric field. Near the particle contact points, the field intensity and the number of high-energy electrons increase.

From the voltage and current waveforms of the CO_2_ discharge, changes of the discharge characteristics in the reactor with and without the packing materials can be seen. A typical filamentous discharge can be seen in the empty tube, which can be verified by a plurality of burr peaks in the current signal of Figure 4a. On the contrary, as is exhibited in Figure 4b–d, filling the foam metals into the discharge region produces a typical packed-bed effect and results in the discharge characteristics to change from a filamentary discharge to a combination of surface discharge and filamentary discharge, because the spike discharge signal is reduced as demonstrated. In the packed bed DBD reactor, the filamentary discharge can only be emerged in a small gap between the particle-particle and the particle-tube walls, while the surface discharge can be generated on the particle surfaces near the particles.

### 3.2. Effect of Packing Materials on CO_2_ Conversion and CO Yield

The CO_2_ conversions and CO yield when different filling foam metals were placed in the discharge gap are displayed in Figure 5. Clearly, packed bed DBD that was filled with three foam metals shows higher CO_2_ conversion and CO yield than DBD without packing. Notably, CO_2_ conversion and CO yield are greatly increased in the foam Ti-packed reactor, which are approximately three times more than that of the CO_2_ conversion and CO yield in the non-packed tube. In the foam Fe-packed tube, CO_2_ conversion and CO yield are lower than the foam Ti-filled tube, but higher than the foam Al-packed tube.

### 3.3. Effect of Discharge Power and Gas Flow Rate on CO_2_ Conversion

The influence of discharge power and gas flow velocity on CO_2_ decomposition is depicted in Figure 6. One of the critical factors affecting the CO_2_ conversion by DBD plasma is the discharge power. The discharge can change the electric field of the plasma reactor, thereby changing the amount of active electrons and active radicals [29]. For an empty tube, it can be noted that increasing the discharge power from 30 W to 90 W results in an improvement in CO_2_ conversion. However, it drops as the discharge power is further increased to 110 W. When the flow rate is fixed at 20 mL/min, the empty tube reaches a maximum conversion of 21.15% at a power of 90 W. The CO_2_ conversion tends to be saturated and slightly decreased if the discharge power is continuously increased over 90 W, and 90 W may be the optimum power under this condition, so from the viewpoint of energy-saving, an appropriate discharge power range is required. Similarly, for foam Fe, when the flow rate is 20 mL/min, the maximum conversion rate of 44.84% is achieved at a power of about 70 W. However, further increases in discharge power have no obvious impact on CO_2_ conversion, because the discharge filaments cover the electrode surface and the quantity of microdischarges is not obviously increased [30].

However, for foam Al and Ti, increasing the discharge power increases CO_2_ conversion. As the plasma power changes from 30 to 110 W, CO_2_ conversion increases from 26.58% and 30.16% to 44.02% and 46.61%, respectively. Increasing the plasma discharge power will increase the amplitude and number of current pulses, as depicted in Figure 7. A higher input power means more energy is put into the reaction system, which leads to a higher current density, producing more cation CO^2+^ and anion O^2−^ in DBD plasma [31,32,33]. This phenomenon indicates that, as the discharge power increases, the quantity of microdischarges are effectively increased, which suggests that more chemical reaction channels and chemically active species are generated for CO_2_ processing [22,34]. Sufficient energy will activate the electrons and reactant molecules to increase the chances of mutual collision frequency between the active materials, so more chemical bonds will be destroyed and more active species will be produced, resulting in a higher CO_2_ conversion [32,35].

It has been proved that the feed flow velocity is also one of the important factors affecting the CO_2_ conversion. It can be noticed from Figure 7 that the CO_2_ conversion rate is the largest at the minimum CO_2_ flow rate of 20 mL/min, whether it is an empty tube or a filling foam metal tube. Similar phenomena have been reported in a non-packed DBD reactor [22]. When the other parameters are not changed, enhancing the CO_2_ flow rate decreases the residence time in the discharge area, which reduces the possibility of CO_2_ conversion caused by collision with high-energy electrons and chemically reactive species, and thus reduces the CO_2_ conversion rate. For example, when the CO_2_ flow velocity varies within a range of 20 to 100 mL/min, CO_2_ conversion is reduced from 46.61% to 33.98% at 110 W for foam Ti.

### 3.4. Effect of Discharge Power and Gas Flow Rate on Energy Efficiency

Energy efficiency is a big concern in plasma-assisted CO_2_ decomposition processes. Among various plasmas, microwave plasma decomposition can achieve higher energy efficiency, owing to the selective excitation of the vibration level of CO_2_ [36]. However, the energy efficiency of CO_2_ splitting is generally around 10% in DBD plasma, because the active particle impact of CO_2_ is a vital procedure in the plasma [12,37].

Figure 8 denotes the influencing factors of energy efficiency, namely, discharge power and gas flow velocity. The energy efficiency in filled foam metal tubes is higher than that of empty tubes over the input power range of 30 to 110 W that was studied. From the figure, we can conclude that energy efficiency drops with increasing discharge power. For foam Ti, the higher the power, the higher the CO_2_ conversion; the energy efficiency decreased from 8.85% to 5.96%, while the plasma discharge power is varied from 30 to 110 W.

The influence of feed flow velocity and discharge power on the energy efficiency of CO_2_ splitting by plasma exhibits the opposite behavior. A higher feed flow rate results in lower CO_2_ conversion, but more energy efficient. For example, by increasing the velocity of feed flow from 20 to 100 mL/min, the energy efficiency increases from 3.88% to 8.85%. Moreover, the reason why energy efficiency decreases with increasing discharge power may be attributed to energy loss, which is released in the form of heat, and it can be verified from an increase in condensate temperature. Under the same plasma operating conditions, the CO_2_ conversion and energy efficiency cannot simultaneously be taken into account during discharge process. Thus, there is a need to balance the relationship between CO_2_ decomposition and energy efficiency during the experiment.

### 3.5. Effect of SIE on CO_2_ Conversion and Energy Efficiency

According to the definition of SIE (Equation (4)), the variation of SIE is determined by the feed flow velocity and discharge power. Generally speaking, SIE is regarded as one of the important individual factors for CO_2_ transformation process [6,38,39]. Obviously, CO_2_ conversion and energy efficiency have opposite trends with changes in SIE. As a consequence, there is a demand to investigate the impact of the optimal SIE calculated by different combinations on the CO_2_ transformation process [40].

Figure 9a shows the impact of SIE on CO_2_ conversion. The introduction of foam metal catalysts into the DBD reactor has a significant effect on the CO_2_ decomposition rate, which increases with increasing SIE. Compared with the results of without packing, the foam Fe, Al, and Ti prominently raise CO_2_ conversion by 2.14, 2.42, and 2.63 times, respectively. Among the three catalysts, foam Ti catalyst has the best catalytic effect.

The effect of SIE on energy efficiency is shown in Figure 9b. The energy efficiency of all the reactors decreases with increasing SIE [41]. For instance, for foam Ti, when the SIE is increased from 42 to 210 kJ/L, the energy efficiency is decreased from 7.11% to 2.33%. Therefore, the interaction effects of power and CO_2_ flow velocity should be taken into account while using a suitable SIE to achieve high CO_2_ conversion and energy efficiency simultaneously.

### 3.6. Effect of Different Packing Materials on CO Selectivity

The influence of different packed foam metals on CO selectivity at different discharge powers is exhibited in Figure 10. For foam Fe and Al, the CO selectivity during the whole reaction process is maintained at about 88–96%. Compared with the empty tube discharge, the difference in CO selectivity between the two at different discharge powers is not significant under our experimental conditions. It can be inferred that, for foam Fe and Al, discharge power plays a secondary role in CO selectivity. However, for foam Ti, the CO selectivity varies between 91% and 97%, which is much higher than the other two foam metals, indicating that the foam Ti has a good catalytic effect. The CO selectivity based on carbon atoms (Equation (3)) is close to 100%, which means that the stoichiometric conversion of CO_2_ to CO has been achieved in this study, and the generation of CO is mainly due to CO_2_ dissociation. The electron collision dissociation of CO_2_ is likely to lead to ground state CO and both the ground state and metastable O atoms, whereas studies have revealed that CO bands are observed in experiments, indicating that CO can also be formed in the excited state [42].

### 3.7. DFT Validations

A prerequisite for highly efficient catalysts is the adsorption of CO_2_ on the catalysts’ surface. To gain a clear insight into the adsorption performance of catalysts, an in-depth investigation of the CO_2_ adsorption was carried out. According to X-ray diffraction (XRD) analysis, the (101) plane of Ti, (111) plane of Al, and (110) plane of Fe are investigated carefully. Based on the above results, we performed DFT calculations to investigate the adsorption of CO_2_ on catalysts’ surface as shown in Figure 11. The results indicate that the adsorption energy of CO_2_ on the (101) plane of Ti is −2.49 eV (Figure 11a), which is lower than that observed for the (111) plane of Al (Figure 11b) and (110) plane of Fe (Figure 11c). This indicates that CO_2_ would have more favorable adsorption on the (101) plane of Ti rather than the (111) plane of Al and (110) plane of Fe. These findings explain the origin of the high catalytic activity of the (101) plane of Ti. Accordingly, the (101) plane of Ti is superior to the (111) plane of Al and (110) plane of Fe for the CO_2_ conversion process.

## 4. Conclusions

In this study, we investigated a coaxial DBD plasma junctions and different packed foam metal materials (foam Fe, Al, and Ti) placed into the discharge gap for CO_2_ conversion. The effects of power, flow velocity, and other discharge characteristics were studied to better understand the influence of the filling catalysts on CO_2_ decomposition. The filling of foam metals in the reactor caused changes in discharge characteristics and discharge patterns, from the original filamentary discharge to both the current filamentary discharge and surface discharge. Compared with the maximum CO_2_ conversion of 21.15% and energy efficiency of 3.92% in the reaction tube without the foam metal electrode, a maximum CO_2_ decomposition rate of 44.84%, 44.02%, and 46.61% and energy efficiency of 6.86%, 6.19%, and 8.85% were obtained in the reaction tubes packed with foam Fe, Al, and Ti, respectively. The results showed that the CO_2_ conversion rate for the filled foam metal electrode was obviously enhanced when compared with the non-packed tube. It was seen that the foam Ti had the best CO_2_ decomposition rate among the three foam metals.

Furthermore, we used density functional theory (DFT) to further verify the experimental results. According to X-ray diffraction (XRD) analysis, the (101) plane of Ti, (111) plane of Al, and (110) plane of Fe were carefully investigated. Based on the above results, we performed DFT calculations to investigate the adsorption of CO_2_ on catalysts surface. The results indicated that the adsorption energy of CO_2_ on the (101) plane of Ti was −2.49 eV, which was lower than the (111) plane of Al and (110) plane of Fe. The results showed that CO_2_ adsorption had a lower activation energy barrier on the foam Ti surface. The theoretical calculation is consistent with the experimental results, which better explains the mechanism of CO_2_ decomposition.

## Figures and Tables

**Figure 1 nanomaterials-09-01595-f001:**
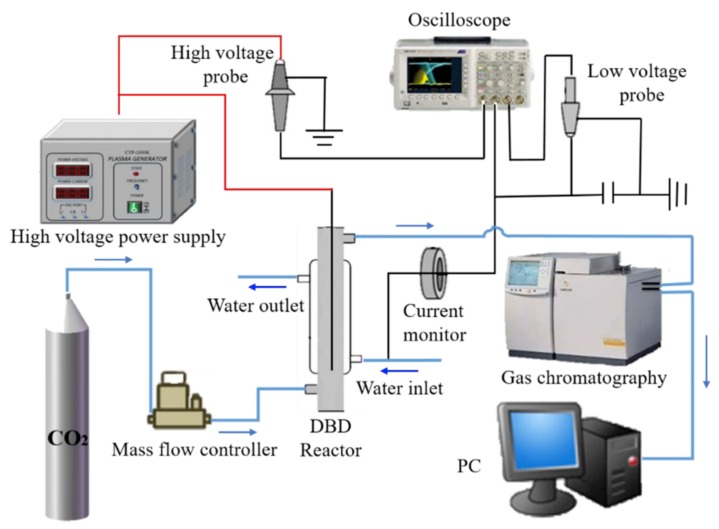
Schematic diagram of the experimental device for CO_2_ splitting.

**Figure 2 nanomaterials-09-01595-f002:**
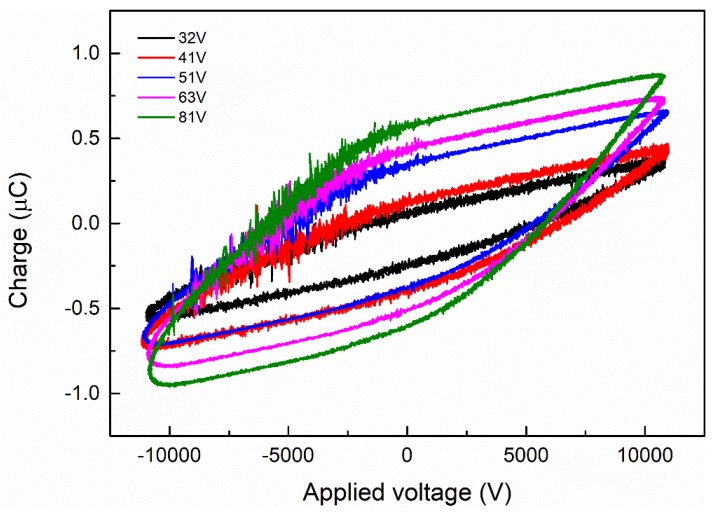
Lissajous diagram of a dielectric barrier discharge (DBD) reactor at different voltages.

**Figure 3 nanomaterials-09-01595-f003:**
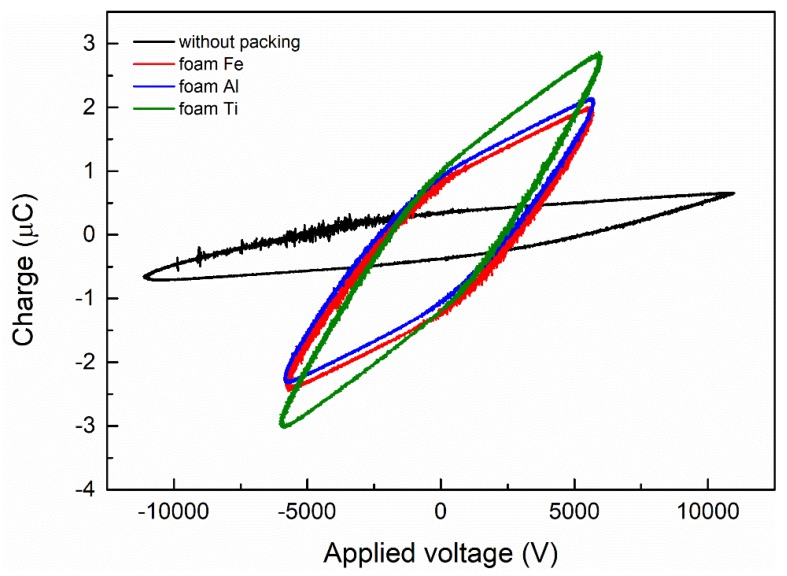
Lissajous figure of CO_2_ decomposition in the DBD reactor with or without foam metals.

**Figure 4 nanomaterials-09-01595-f004:**
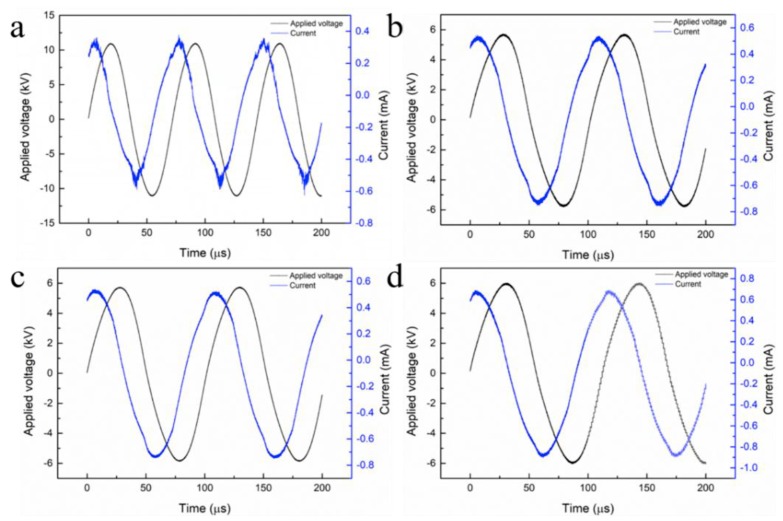
Discharge characteristics of CO_2_ in the DBD reactor without or with filling foam metals: (**a**) Without packing; (**b**) Foam Fe; (**c**) Foam Al; (**d**) Foam Ti.

**Figure 5 nanomaterials-09-01595-f005:**
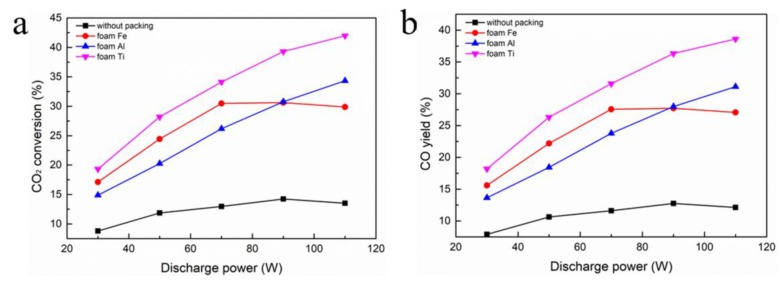
Effect of packing materials on: (**a**) CO_2_ conversion; (**b**) CO yield.

**Figure 6 nanomaterials-09-01595-f006:**
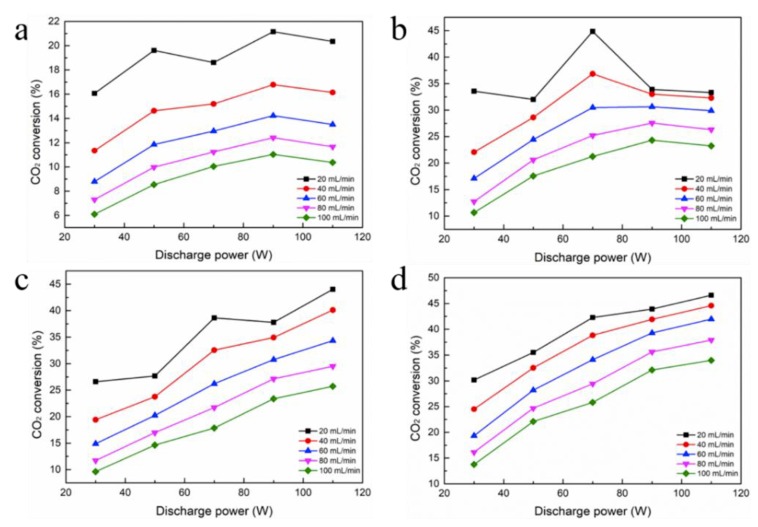
Effect of discharge power and gas flow velocity on CO_2_ conversion: (**a**) Without packing; (**b**) Foam Fe; (**c**) Foam Al; (**d**) Foam Ti.

**Figure 7 nanomaterials-09-01595-f007:**
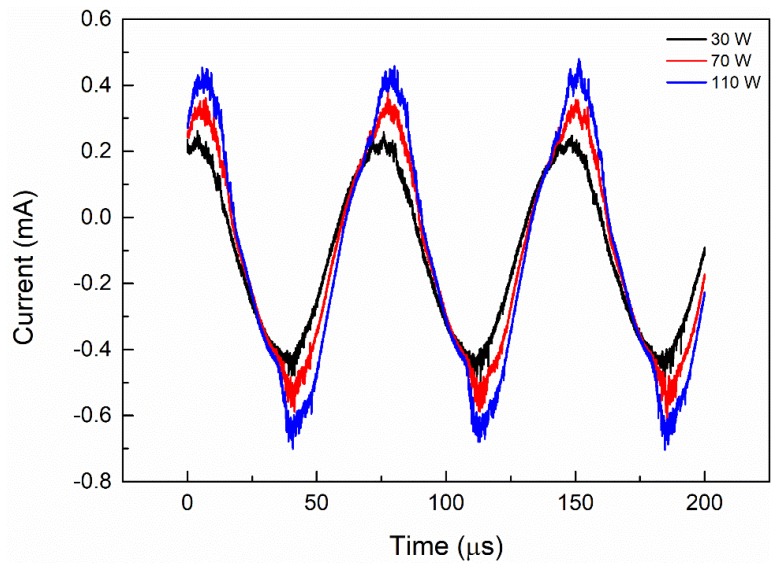
Effect of discharge power on the current signals for empty tube.

**Figure 8 nanomaterials-09-01595-f008:**
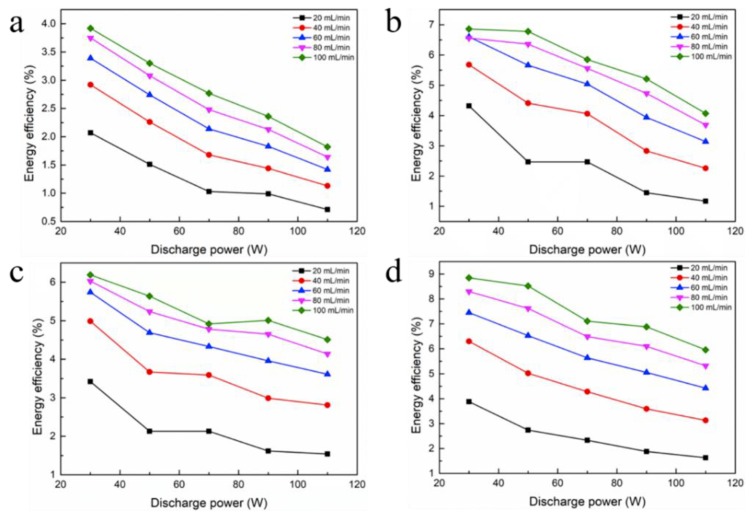
Effect of discharge power and gas flow velocity on energy efficiency: (**a**) Without packing; (**b**) Foam Fe; (**c**) Foam Al; (**d**) Foam Ti.

**Figure 9 nanomaterials-09-01595-f009:**
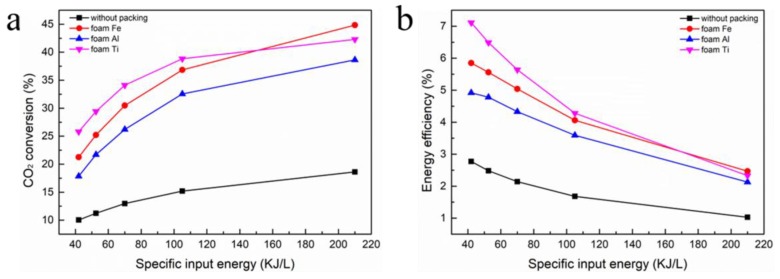
Effect of specific input energy of different foam metals on: (**a**) CO_2_ conversion; (**b**) energy efficiency.

**Figure 10 nanomaterials-09-01595-f010:**
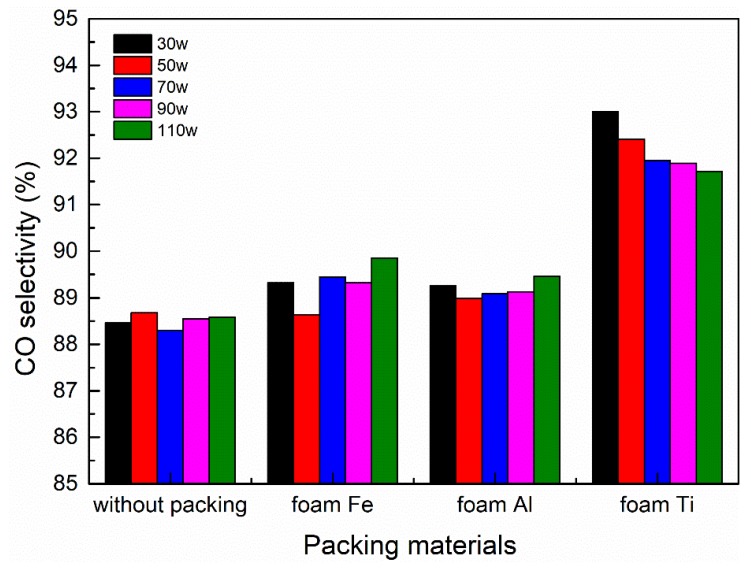
CO selectivity of different foam metals at different discharge powers.

**Figure 11 nanomaterials-09-01595-f011:**
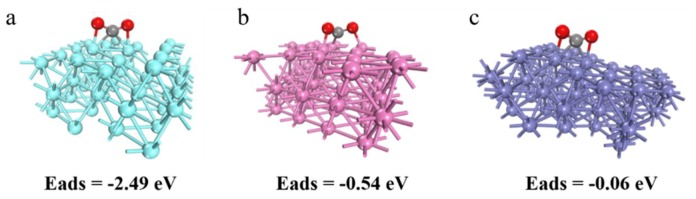
Side views of the optimized geometric structures of CO_2_ adsorption on: (**a**) (101) plane of Ti; (**b**) (111) plane of Al; (**c**) (110) plane of Fe. Gray and red balls stand for the C and O atoms, respectively.

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
