# Peer review of "DBD Plasma Combined with Different Foam Metal Electrodes for CO2 Decomposition: Experimental Results and DFT Validations"

_nanomaterials, 2019, doi:10.3390/nano9111595_

Round 1

Reviewer 1 Report

The paper is well written, the idea is clear. The experimental results seem to correct. Unfortunately, paper needs a major revision before publication. The main unclear point is the follow: what is the difference between Ti and Fe (or Al) electrodes? Performed DFT calculations are not explaining the difference in CO2 conversion, or energy effectiveness.

I’d like to formulate some specific questions:

What is the surface area of each electrode? Is it possible, that change in selectivity is the function of surface area, but not of the foam material? What is the crystal structure of electrodes? Pore size distribution or initial crystals size? Do the foam electrodes have any impurities? Small amount of other material could affect the CO2 adsorption or decomposition. Do the reaction conditions affect on electrodes structure (surface area, pore size distribution, weight loss, surface carbonization of oxidation)? Do the electrode material (or gas flow) warming up under reaction conditions? If yes, how it correlates with CO2 conversion and energy efficiency? On figure 5 on can see the saturation of CO2 conversion above 70W in case of Fe foam electrode. Do the other materials reach the same “plateau” at higher power? Do the weight or size of electrode affect on maximum conversion of CO2 or energy effectiveness? What is the composition of coproducts? Is it a amorphous carbon on the electrode surface? Or CxHyOz organic compaunds? The maximum of CO2 conversion was observer at lower flow rate. Wat is the conversion value in case of static reactor conditions (of in case of use of 50:50 CO+CO2 mixture in the enter of reactor)? Plasma decomposition is nonequilibrium conditions process, thus where is no CO2/CO equilibrium reaction limitation. 100% CO2 conversion could be reached in theory. Why it is not so? What facts are demonstrating DFT calculations? Why such crystals planes were used? In the paper where is no any experimental facts about surface structure of foam electrodes. I doubt, that adsorption energy on ideal planes have any correlations with CO2 conversion, observed in the experiments.

Author Response

Response to Reviewer 1 Comments

Dear Editors and Reviewers,

We greatly appreciate the time and effort you’ve spent in reviewing our manuscript (nanomaterials-593845) and your constructive suggestions are valuable to our manuscript and future research work. We have carefully considered the comments and suggestions from reviewers. We hope the editor and reviewers will be satisfied with the revisions for the original manuscript. The followings are the answers and comments to reviewers’ questions and suggestions. And changed portion have been highlighted by red text in the revised manuscript.

Point 1: What is the surface area of each electrode? Is it possible, that change in selectivity is the function of surface area, but not of the foam material?

Response 1: Thanks for the advice. The three materials we use are foam Fe, foam Al and foam Ti, in which foam Fe and foam Al are a kind of network structure with loose and porous inside. The structure of foam Ti is similar to that of steel plate with compact surface. We are sorry that we did not further verify it through BET, but it can be observed by naked eye that the surface area of the first two foam metals is much larger than that of foam Ti. However, the result is that foam Ti works best, so there is reason to believe that selectivity should be more related to the type of material.

          Foam Fe                                   Foam Al                                   Foam Ti

Point 2: What is the crystal structure of electrodes?

Response 2: Thanks for your pointing out the question.

According to the XRD analysis, the foam Fe belongs to the Im-3m space group, a=b=c=2.861, α=β=γ=90.00°; The theoretical model parameters of foam Fe also belong to the Im-3m space group,a=b=c=2.866, α=β =γ= 90.00 °;

Foam Al belongs to the Fm-3m space group, a=b=c=4.050, α=β=γ=90.00°; The theoretical model parameters of foam Al also belong to the Fm-3m space group,a=b=c=4.050, α=β =γ= 90.00 °;

Foam Ti belongs to the P63/mmc space group, a=b=2.920,c=4.670, α=β=90°ï¼Œγ=120.00°ï¼›The theoretical model parameters of foam Ti also belong to the P63/mmc space group,a=b=2.951,c=4.679, α=β=90.00 °ï¼Œγ=120.00 °;  

The experimental results are consistent with the theory, which indicates the feasibility of selecting the theoretical model.

Point 3: Pore size distribution or initial crystals size?

Response 3: Thanks for your pointing out the problem. The initial crystal is a whole piece of plate size 30 cm long and 20 cm wide. We cut it into a granular or rectangular shape with a diameter of about 5 mm by hand cutting,so that it can be filled into the reaction tube, where the filling volume remains the same.

Point 4: Do the foam electrodes have any impurities? Small amount of other material could affect the CO2 adsorption or decomposition.

Response 4: Thanks for your careful attention. The foam electrode we use is pure elemental metal with no impurities.

Point 5: Do the reaction conditions affect on electrodes structure (surface area, pore size distribution, weight loss, surface carbonization of oxidation)?

Response 5: Thanks for your question. The reaction conditions such as discharge power, flow rate and circulating water temperature all have an effect on the decomposition efficiency of CO2. This experiment, we used foam metals to fill the discharge gap. It is found that porous foam Fe, Al and Ti not only played a role as carrier of energy transformation and electrode distributed in discharge gaps but also promoted the equilibrium shifting toward the product side to yield more CO by consuming some part of O2 and O radicals generated from the decomposition of CO2. By promoting the reaction to move forward, the surface of the foam metal will be oxidized and carbonized. So the reaction conditions will affect the surface oxidation and carbonization of the electrode structure.

Point 6: Do the electrode material (or gas flow) warming up under reaction conditions? If yes, how it correlates with CO2 conversion and energy efficiency?

Response 6: Thanks for your careful attention. In this experiment, we used a self-cooling DBD reactor with externally circulated water. When the circulating water is not added, we have previously measured the temperature inside the tube by infrared thermometer, which is about 200~300 °C. With the increase of reaction time, the internal electrode material will definitely rise in temperature. Our previous studies have shown that when the circulating water temperature rises, the conversion rate and energy efficiency are reduced to different extents. Meanwhile, the conversion rate and energy efficiency are highest when the circulating water is fed into 20°C. Therefore, we use self-cooling DBD reaction device, and the outside of the reaction tube is circulated with water at 20 °C to keep the temperature inside the tube as high as about 20 °C.

Point 7: On figure 5 on can see the saturation of CO2 conversion above 70W in case of Fe foam electrode. Do the other materials reach the same “plateau” at higher power?

Response 7: Thanks for your careful attention. Similar studies were conducted on foam Ni and foam Cu before. The results show that the CO2 conversion rate of the two foam electrodes increases with the increase of discharge power. However, the above “plateau” phenomenon did not occur. We suspect that foam Fe is different from other foam metal structures, and its adsorption site near 50W is saturated, while the adsorption sites of other metal foams have not reached saturation state in this state. Therefore, as power increases, the conversion rate also increases.

Point 8: Do the weight or size of electrode affect on maximum conversion of CO2 or energy effectiveness?

Response 8: Thanks for your question. We have previously done research of the effect of ZrO2 pellets as a filling material on CO2 conversion rate and energy efficiency. The average diameter of the pellets was 1.1 mm, 1.5 mm and 1.9 mm, respectively.  The experimental results show that the smaller the diameter of the ZrO2 sphere, the higher the conversion and energy efficiency. We think that the smaller the size of the metal foam electrode, the larger the specific surface area, and the more exposed active sites, the better the effect should be. At the same time, the smaller the size, the electric field distribution near the contact points is stronger, and the higher the energy density, the reaction effect is better.

Point 9: What is the composition of coproducts? Is it a amorphous carbon on the electrode surface? Or CxHyOz organic compaunds?

Response 9: Thanks for your question. During the experiment, only pure CO2 was injected. We found that there was black carbon deposition in the Teflon tube after the reaction. SEM and EDX analysis in previous studies also proved that there was carbon deposition on the catalyst, so the by-products should be carbon black.

Point 10: The maximum of CO2 conversion was observer at lower flow rate. What is the conversion value in case of static reactor conditions (of in case of use of 50:50 CO+CO2 mixture in the enter of reactor)?

Response 10: Thanks for your pointing out the question. At a low flow rate, the residence time of CO2 in the reaction tube is longer, and the probability of collision with the active particles is larger, so the conversion rate is higher. We are very grateful for your proposal to adopt 1:1 feeding of CO and CO2 mixture at the reactor inlet. We are sorry to say that we do not conduct the experiment in this case. We hope to conduct experimental in subsequent research.

Point 11: Plasma decomposition is nonequilibrium conditions process, thus where is no CO2/CO equilibrium reaction limitation. 100% CO2 conversion could be reached in theory. Why it is not so?

Response 11: Thanks for your question. We infer that there may be two cases: First, during the process of the non-equilibrium plasma decomposition conditions,the C-O bond of CO2 is dissociated by the active material, but the reaction of CO2 into CO and O2 is a reversible reaction. A part of the active CO radicals and O radicals are recombined to move in the reverse reaction direction, thereby reducing the CO2 conversion rate. Second, since the reaction is carried out in the mobile phase, some of the CO2 may have gone out before the reaction. Therefore, the theoretical 100% CO2 conversion rate is not achieved.

Point 12: What facts are demonstrating DFT calculations? Why such crystals planes were used? In the paper where is no any experimental facts about surface structure of foam electrodes. I doubt, that adsorption energy on ideal planes have any correlations with CO2 conversion, observed in the experiments.

Response 12: Thanks for your question. According to the XRD analysis,  the crystal planes exposed by the foams Fe, Al and Ti are 110 plane, 111 plane and 101 plane, respectively. Therefore, we choose these crystal planes for theoretical calculations. Although the DFT calculation can not directly explain the improvement of CO2 conversion and reflect the performance of the catalyst, the first step of the catalytic reaction is gas adsorption. The gas that is more easily adsorbed preferentially adsorbs on the surface of the catalyst. When the adsorption energy is large, it is easier to adsorb the gas and cause potential activation of the CO2 molecule. Moreover, the theoretical calculation ignores many external influence factors and may have some deviations from the experimental results.

Reviewer 2 Report

Comments on

 DBD Plasma Combined with Different Foam Metal Electrodes for CO2 Decomposition: Experimental Results and DFT Validations

The authors of this paper presented an interesting research about nonthermal plasma. The paper is accepted for publication if the authors reply to reviewer’s comments and modify the paper accordingly:

1) More detailed figure of DBD reactor is necessary.

2) What is the originality of the paper? The authors should explain this in the introduction part.

3) The paper should be checked by a native speaker. There are several sentences which need to be corrected. Examples:

Page1, line 40:

“Recently, people have conducted extensive researches on the impacts of carbon dioxide (CO2).”

Page 2, line 88

“The catalysts used by the above researchers are all metal oxides, but little or no research on the 88 conductive foam metal elements”.

4) CO selectivity was obtained around 89 to 93 %, results were pretty good with such high selectivity, but the other 11 to7% was converted to what?

This kind of plasma chemical reactions, authors should mention the byproduct analysis which usually GCMS is utilized to analyze. Especially for discussion the catalytic effect of metal materials. In this experimental work, only GC analysis is not enough, and add some more references to show the byproduct analysis or reaction.

Author Response

Response to Reviewer 2 Comments

Dear Editors and Reviewers,

We greatly appreciate the time and effort you’ve spent in reviewing our manuscript (nanomaterials-593845) and your constructive suggestions are valuable to our manuscript and future research work. We have carefully considered the comments and suggestions from reviewers. We hope the editor and reviewers will be satisfied with the revisions for the original manuscript. The followings are the answers and comments to reviewers’ questions and suggestions. And changed portion have been highlighted by red text in the revised manuscript.

Point 1: More detailed figure of DBD reactor is necessary.

Response 1: Thanks for your advice. The detailed figure of DBD reactor is shown below:

Point 2: What is the originality of the paper? The authors should explain this in the introduction part.

Response 2: Thanks for your pointing out the question. The originality of this paper is to use foam metal elemental for CO2 decomposition reaction. Currently, there are few researches about foam metal for CO2 decomposition. Compared with the traditional metal oxide catalyst, the foam metal element can react with the O radicals in the reaction tube and consume O2 and O radicals, which break the balance, promoting the CO2 decomposition.

At present, in the literatures we have seen, most of the catalysts used for CO2 decomposition by plasma are metal oxides. However, few literatures have reported researches on CO2 conversion by using foam metal as catalyst. So in this work, we studied DBD plasma junctions and different packed foam metal materials (Fe, Al, Ti) for CO2 conversion. It can be found that the foam metals can not only serve as a carrier for energy transformation, but also the foam metals distributed in the discharge gap consumes a part of O2 and O radicals, so that the reaction proceeds in the positive direction, promoting the decomposition of CO2. The effects of power, flow velocity and other discharge characteristics are studied to better understand the influence of the filling catalysts on CO2 splitting. Further more, we use density functional theory (DFT) to further verify the experimental results. The results show that CO2 adsorption has a lower activation energy barrier on the foam Ti surface.

Point 3: The paper should be checked by a native speaker. There are several sentences which need to be corrected. Examples:

Response 3: Thanks for your pointing out the error.

Page1, line 40: Recently, the environmental impact of carbon dioxide (CO2) has been extensively studied.

Page 2, line 88: At present, in the literatures we have seen, most of the catalysts used for CO2 decomposition by plasma are metal oxides. However, few literatures have reported researches on CO2 conversion by using foam metal as catalyst.

Point 4: CO selectivity was obtained around 89 to 93 %, results were pretty good with such high selectivity, but the other 11 to 7% was converted to what?

Response 4: Thanks for your question. During the experiment, only pure CO2 was injected. We found that there was black carbon deposition in the Teflon tube after the reaction. SEM and EDX analysis in previous studies also proved that there was carbon deposition on the catalyst, so the by-products should be carbon black.

Point 5: This kind of plasma chemical reactions, authors should mention the byproduct analysis which usually GCMS is utilized to analyze. Especially for discussion the catalytic effect of metal materials. In this experimental work, only GC analysis is not enough, and add some more references to show the byproduct analysis or reaction.

Response 5: We are very grateful for your valuable suggestions. In the previous study, we conducted EDX analysis and SEM characterization of the catalyst after the reaction. The results showed that there was black carbon deposition on the catalyst surface after the end of the reaction. Moreover, only CO2 gas is injected during the reaction, so the by-product is carbon black. We will improve the experiment and design a more rigorous experimental scheme in the later research.At the same time, we will combine your proposed GCMS method to further analyze the composition of by-products more accurately.

Round 2

Reviewer 2 Report

The authors replied to the comments so the paper can be published.